# Multi-Stage Multi-Scale Local Feature Fusion for Infrared Small Target Detection

**Yahui Wang** , **Yan Tian \***, **Jijun Liu and Yiping Xu**

School of Electronic Information and Communications, Huazhong University of Science and Technology, Wuhan 430074, China; m202172619@hust.edu.cn (Y.W.); d202180819@hust.edu.cn (J.L.); xuyiping@hust.edu.cn (Y.X.)
**\*** Correspondence: tianyan@hust.edu.cn

**Abstract:** The detection of small infrared targets with dense distributions and large-scale variations is an extremely challenging problem. This paper proposes a multi-stage, multi-scale local feature fusion method for infrared small target detection to address this problem. The method is based on multi-stage and multi-scale local feature fusion. Firstly, considering the significant variation in target sizes, ResNet-18 is utilized to extract image features at different stages. Then, for each stage, multi-scale feature pyramids are employed to obtain corresponding multi-scale local features. Secondly, to enhance the detection rate of densely distributed targets, the multi-stage and multi-scale features are progressively fused and concatenated to form the final fusion results. Finally, the fusion results are fed into the target detector for detection. The experimental results for the SIRST and MDFA demonstrate that the proposed method effectively improves the performance of infrared small target detection. The proposed method achieved mIoU values of 63.43% and 46.29% on two datasets, along with F-measure values of 77.62% and 63.28%, respectively.

**Keywords:** infrared small target; multi-scale local feature pyramid; multi-scale feature fusion



## 1. Introduction

Object detection is a crucial technique in computer vision [1], which involves identifying and localizing the objects of interest in an image. Object detection is a fundamental task in computer vision, serving as the foundation for many advanced visual tasks such as autonomous driving [2], pedestrian detection [3], and face recognition [4]. These tasks rely on object detection to determine the positions of objects within images or videos. Object detection is a pivotal step in image understanding, enabling computers to recognize and comprehend the presence, location, and shape of various objects. It stands as a core challenge within the field of computer vision, playing a vital role across a diverse range of applications. Moreover, with ongoing technological advancements, the research on object detection continues to be a prominent and evolving focus within the field of computer vision.

Traditional object detection algorithms typically involve image preprocessing, candidate region generation, feature extraction, feature classification, bounding box adjustment, and postprocessing processes. The purpose of image preprocessing is to preprocess the input image in order to enhance the effectiveness during the subsequent processing. This involves tasks such as image scaling, grayscale conversion, contrast enhancement, noise removal, and so on. Object candidate region generation is the process of generating a set of candidate regions in an image that potentially contain the target objects. The common method is the sliding windows method. The generated candidate regions can be bounding boxes of different sizes and shapes or image regions. Feature extraction involves extracting features from each candidate region. The traditional feature extraction methods include the use of color histograms, texture features, shape features, edge features, and so on. These

features can be used to describe the visual attributes of candidate regions. Feature classification involves using a classifier to categorize each candidate region and determine whether it contains the target object. The commonly used classifiers include support vector machine (SVM) [5], AdaBoost [6], and random forest [7] classifiers. The bounding box adjustment is typically performed on candidate regions classified as targets to more accurately enclose the target object. This involves adjusting the position, size, and shape of the bounding box. The postprocessing is performed on the results of the classification and bounding box adjustment phases. This includes removing overlapping candidate boxes, filtering and sorting the results based on certain rules or criteria, and ultimately generating the final output of the object detection results.

Due to the advantages of all-weather detection, covert detection, and long-distance detection offered by infrared imaging systems [8], infrared small target detection technology has been widely applied in fields such as early warning, guidance, and surveillance systems [9,10]. However, there are several challenges in practical applications. Firstly, due to the long imaging distance, infrared small targets usually occupy only a small number of pixels in the image. Secondly, the energy loss occurs during the propagation process, resulting in relatively low grayscale values for infrared targets. These two factors result in a lack of shape, texture, and color information for infrared small targets [11]. Additionally, the detection results of infrared small targets are easily affected by complex backgrounds and random noise. These aforementioned issues make infrared small target detection a challenging task. Therefore, researching infrared small target detection methods with high detection rates and low false alarm rates holds significant research value and application prospects.

Traditional infrared small target detection methods are typically constrained by manually designed feature extraction methods. Manually designed feature extraction algorithms come with several limitations. They are unable to fully capture the rich features of targets. As the target features and backgrounds change, manually designed feature extraction methods become ineffective. Additionally, due to noise and the influence of strong background edges, manually designed feature extraction methods often erroneously focus on noise and strong background edges. These manually designed methods also lack practicality, as they need to be adjusted when the application scenario changes. Therefore, novel approaches are required. To overcome these limitations, researchers have in recent years turned to advanced techniques and started exploring the use of generic object detectors for infrared small target detection. However, they have encountered challenges due to the weak features of infrared small targets and their vulnerability to complex backgrounds. As a result, directly applying generic object detectors for infrared small target detection has proven to be ineffective. As shown in Figure 1, there are often issues such as imprecise bounding boxes, target losses, and high false alarm rates. These problems can arise due to various factors, including the complexity of the background, low contrast of infrared small targets, noise, and variations in target appearance. In order to address the challenge of adapting generic object detectors for infrared small target detection, researchers began treating the problem of infrared small target detection as a binary classification task. They designed intricate semantic segmentation networks [12–14] to segment infrared small targets from complex backgrounds, thereby achieving infrared small target detection. However, this approach proves ineffective in scenarios characterized by significant variations in target size and high target densities. The method proposed in this paper builds upon the existing research.

To address these issues, this paper proposes an infrared small target detection method based on multi-stage and multi-scale local feature fusion. Our method consists of three components: a backbone network, three multi-scale local feature pyramid modules, and a multi-scale feature fusion module.

The backbone network serves as the foundation of our method and is responsible for extracting high-level features from the input image. The multi-scale local feature pyramid module aims to capture local details and contextual information at different scales. The multi-scale feature fusion module combines the features from different scales [15] and integrates them into a unified representation.

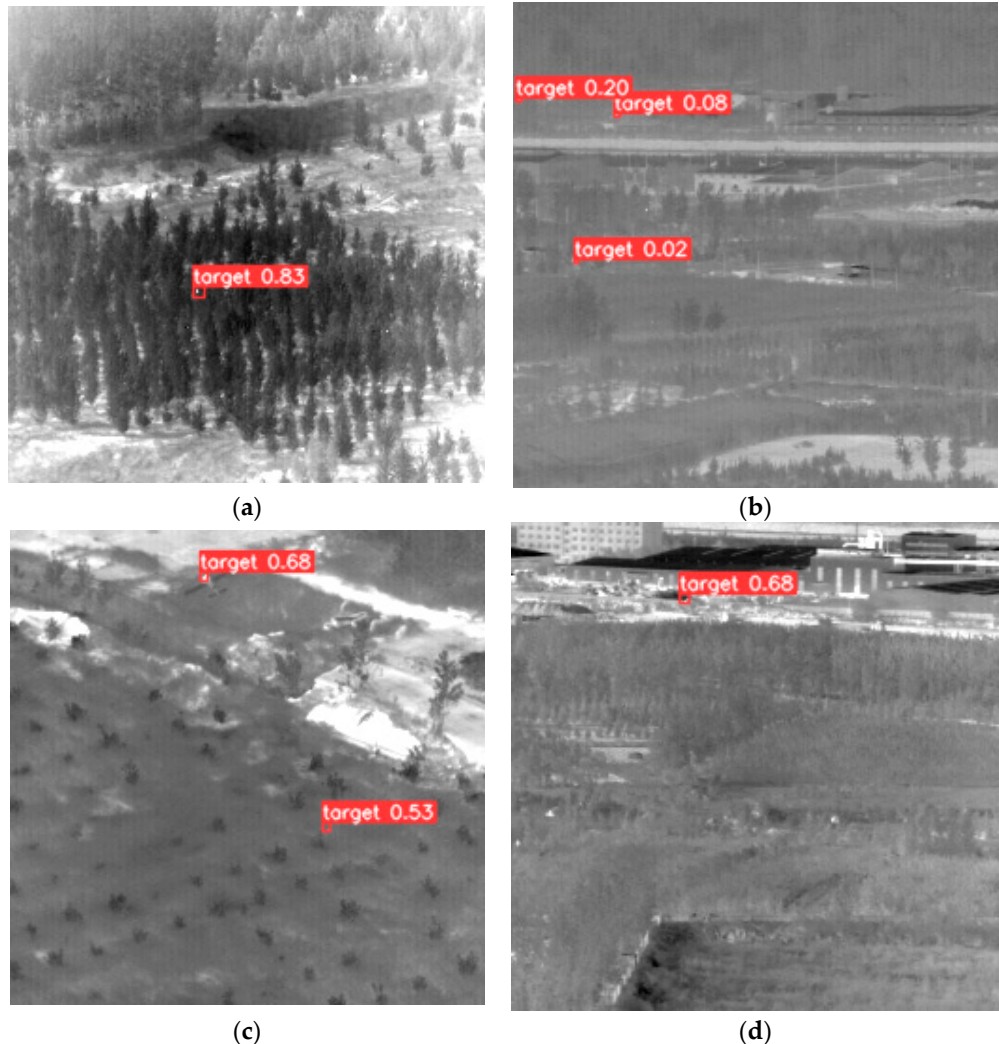

**Figure 1.** Performance of generic object detectors: (**a**) imprecise bounding boxes; (**b**) missed detection and false alarm; (**c**) false alarm; (**d**) missed detection and false alarm.

The contributions of this paper can be summarized as follows: (1) To tackle the problem of existing models being unable to handle multi-scale targets, this paper introduces a multi-scale local feature pyramid module (MLFPM). This module combines local context, local contrast, and local attention mechanisms to extract semantic information from different layers of feature maps. (2) To address the problem of undistinguishable dense small targets, a multi-scale feature fusion module (MFFM) is designed to leverage both deep and shallow features for precise target localization. (3) To validate the effectiveness of the proposed model, the paper conducts ablation experiments on the MDFA [16] and SIRST [17] datasets to evaluate the individual modules. The experimental results demonstrate the effectiveness of each proposed module. Comparisons with state-of-the-art algorithms are also conducted, and the results show that the proposed network achieves excellent performance on two datasets.

The remainder of this paper is structured as follows. Section 2 provides an extensive review of the related work and presents a detailed explanation of our proposed method. In Section 3, an analysis of the proposed method against state-of-the-art approaches is conducted on various datasets. Section 4 focuses on discussing the results of the ablation experiments performed to evaluate the individual components and variations of our proposed method. Finally, in Section 5, the conclusions are presented, summarizing the key findings and contributions of our research.

## 2. Materials and Methods

### 2.1. Related Work

### 2.1.1. Generic Object Detection

With the recent advancements in deep learning techniques, the performance of generic object detectors based on deep convolutional neural networks (DCNNs) has significantly improved. The mainstream generic object detection methods can be broadly categorized based on the following criteria: two-stage or single-stage object detection, and anchor-free or anchor-based object detection.

Two-stage object detectors, such as Faster-R-CNN [18], initially generate region of interest (RoI) proposals as coarse class-agnostic detection results in the first stage. In the second stage, these detectors extract RoI features and perform refined classification and localization processes. Although two-stage object detectors achieve high detection accuracy rates, their inference speeds are relatively slow. On the other hand, one-stage object detectors, such as the YOLO series [19–21] and SSD [22], directly regress the complete detection results in a single prediction step. One-stage detectors are faster and can achieve real-time inference; however, they tend to have slightly lower accuracy rates compared to two-stage detectors. While the method proposed in this paper comprises multiple processing steps, these steps are aimed at generating the detection results directly in the end, without an RoI generation stage. Therefore, this method belongs to the one-stage object detection method, capable of directly producing detection results in a single forward pass.

### 2.1.2. Infrared Small Target Detection

Traditional infrared small target detection methods can be divided into two main categories: local feature-based methods and non-local feature-based methods. Local feature-based methods assume that there are significant differences in grayscale values between the target pixels and their surrounding pixels, while all background pixels have similar grayscale values. These methods detect infrared small targets by extracting the differences between pixels and their surrounding pixels in infrared images. Some common local feature-based methods include maximum mean–median filtering [23], morphological filtering [24], and local contrast methods [25–28]. Non-local feature-based detection methods assume that there are differences between targets and backgrounds in terms of the frequency bands or linear subspaces. These methods do not focus on the local characteristics of the targets but directly process the entire infrared image to separate the targets from the original image. Several representatives of non-local feature-based methods include transform domain filtering [29,30], sparse representation methods [31,32], and sparse low-rank decomposition methods [33,34]. Traditional infrared small target detection methods typically rely on manually designed methods for feature extraction. Manually designed feature extraction methods have certain limitations. Firstly, they may fail to fully extract the rich features of the targets. Secondly, they are influenced by the subjective judgment and experience of the algorithm designer, which makes them less adaptable to different datasets. Lastly, manually designed methods often require extensive experimentation and manual adjustments to achieve good detection performance.

Recently, with the availability of infrared small target datasets and the development of deep-learning-based object detection methods [18–22], deep-learning-based infrared small target detection methods have gained increasing attention. MDvsFA [16] introduces two generators that focus on addressing missed detections and false alarms, and achieved a balance between the two through training a discriminator. ACM [17] focuses on the fusion of deep and shallow features and includes an asymmetric feature fusion structure, which achieved good detection performance on FPN [35] and U-Net [36,37]. ALCNet [38] integrates traditional local contrast-based methods into deep learning networks. RISTD-Net [39] introduces a feature extraction framework that combines handcrafted features with convolutional neural networks. DNANet [40] introduces a dense nested attention network for reusing and fusing deep and shallow features. AGPCNet [41] involves an attention-guided contextual module to obtain and fuse multi-scale features. Although

these methods have been used to investigate infrared small target detection from different perspectives, there are still several limitations. On one hand, the existing methods fail to handle the extraction of multi-scale targets, since the size of infrared small targets can vary significantly. On the other hand, due to the decrease in resolution during feature extraction, features of dense small targets in deep feature maps tend to merge together, making it challenging to separate the targets during feature fusion and leading to missed detections.

The traditional infrared small target detection methods typically rely on manually designed feature extraction techniques. However, they come with certain limitations, such as their limited adaptability to different datasets and the need for extensive experimentation and parameter adjustments. Despite the significant progress made by deep learning methods in the field of infrared small target detection, they still face several challenges. These challenges include effectively extracting multi-scale targets and separating dense small targets within deep feature maps. Deep learning methods offer the advantage of automatically learning features but demand substantial annotated data and computational resources for support. Therefore, when considering the strengths and weaknesses of both traditional and deep learning approaches, a wider range of choices and development directions can be provided for infrared small target detection. The method proposed in this paper is built upon the foundations of both traditional and deep learning methods, with a specific focus on addressing the issues of multi-scale target extraction and the separation of dense small targets within deep feature maps.

### 2.2. Method

### 2.2.1. Overall Architecture

The network architecture proposed in this paper is shown in Figure 2. Firstly, the infrared image is input into the network, and feature extraction is performed using ResNet-18 as the backbone network [42]. This procedure includes three down-sampling stages, resulting in three different-sized feature maps. Secondly, each of the three feature maps is processed by the corresponding MLFPM to obtain multi-scale features. Then, the MFFM is used to fuse the three feature maps. Finally, the fused feature map is fed into the object detector to obtain the final detection results.

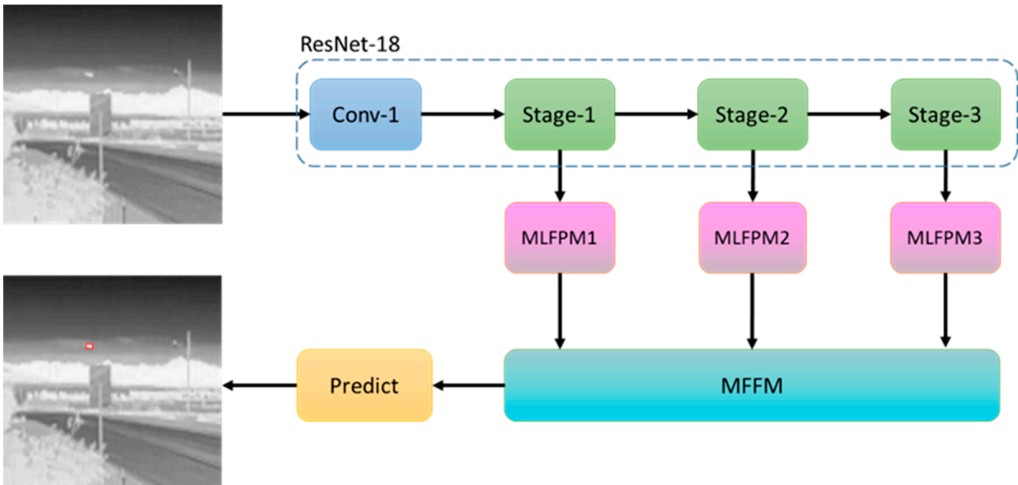

**Figure 2.** Network architecture. Our method consists of three components: a backbone, three multi-scale local feature pyramid modules (MLFPM), and a multi-scale feature fusion module (MFFM). After passing through the backbone network, input images yield feature maps of three layers. These feature maps are then fed into their respective MLFPMs for further extraction. Subsequently, they are passed into the MFFM for feature fusion, ultimately resulting in the output.

ResNet-18 was selected as the backbone network for feature extraction for the following reasons. Firstly, ResNet-18 has already been trained on the large-scale dataset ImageNet [43], which gives its model parameters a good initial state. By fine-tuning the

pretrained weights of ResNet-18 on the infrared dataset, the network can acquire the ability to extract features of infrared images. Furthermore, ResNet-18, being a relatively shallow model among the ResNet series, maintains higher computational efficiency while still achieving a certain level of detection performance. Additionally, for the task of detecting small targets in infrared images, deeper networks tend to focus excessively on global information, disregarding the local details of the targets. Since infrared datasets are typically smaller, deeper networks are more prone to overfitting, while ResNet-18, with fewer model parameters, offers better control over the model complexity and helps mitigate overfitting. The reason for further processing the obtained feature maps of different scales using the corresponding MLFPM is that the target features vary across different stages of the feature maps. The shallow-level feature maps have higher resolutions, rich spatial features, and significant scale variations, while the deep-level feature maps have lower resolutions, rich semantic information, and minor scale variations. A single module in a uniform form would struggle to adapt to the different characteristics of feature maps at different stages. The reason for performing multi-scale feature fusion on the feature maps processed by MLFPM is that the low resolution of the deep-level feature maps causes densely distributed targets to blend together, making it difficult to accurately separate and localize the targets. Meanwhile, the high resolution of the shallow-level feature maps can provide rich spatial information. Through multi-scale feature fusion, the spatial information provided by the shallow-level feature maps and the semantic information provided by the deep-level feature maps are fully utilized. This enables the detection method to better separate densely distributed objects.

### 2.2.2. Multi-Scale Local Feature Pyramid Module

By analyzing the features of the infrared small targets extracted by the backbone network, it was observed that the feature extraction capability of the backbone network varied for targets of different sizes. Larger-sized targets possess richer semantic features in the deep-level feature maps, while smaller-sized targets exhibit weaker semantic features in the deep-level feature maps. This is due to the loss of features during the down-sampling process for smaller-sized targets. Therefore, more attention needs to be given to the shallow-level feature layers. In comparison to deep-level features, shallow-level feature layers contain richer contour information for small targets but have weaker semantic information and are more prone to background interference. Hence, this paper introduces a local feature extraction module to enhance the semantic information for targets in shallow-level feature maps and suppress background interference.

In the previous work, the design of the local feature extraction module employed convolutional pooling, dilated convolution, and attention mechanisms [44–46]. However, this design overlooked the features with high local contrast within the target itself and neglected the contextual information of the target. The proposed local feature extraction module in this paper differs by introducing the local context module and local contrast extraction module and improving the local attention mechanism module. The local context module incorporates the information surrounding the targets, which helps distinguish the targets from complex backgrounds. The local contrast extraction module measures the degree of local brightness changes by calculating the variance and standard deviation of local pixel values, thereby focusing more on features with high local contrast within the target itself. The improved local attention mechanism provides a more detailed and rich representation of local features, assisting in addressing the challenge of separating densely distributed targets in deep-level feature maps. By extracting and fusing features at different scales, the model can comprehensively capture the contextual information and holistic features of the targets, thereby further enhancing the performance during target detection.

The local feature extraction module is shown in Figure 3. It consists of three parts: the local context module, the local contrast extraction module, and the local attention mechanism module.

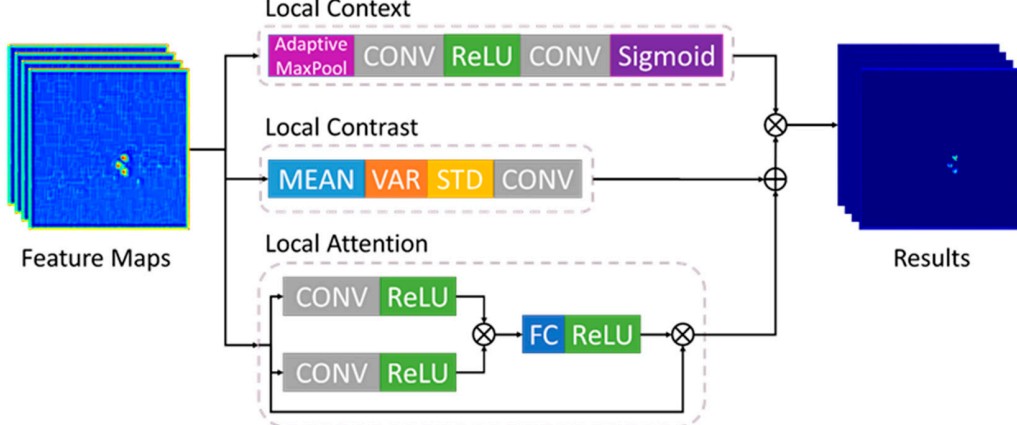

**Figure 3.** Local feature extraction module. The local feature extraction module consists of three parts: a local context module, a local contrast extraction module, and a local attention mechanism module. The local context module consists of an adaptive max-pooling layer, two activation functions, and two convolutional layers. The local contrast extraction module involves calculating the standard deviation of the feature map and then processing it through a convolutional layer. The local attention mechanism module incorporates two convolutional layers, three activation functions, and one fully connected layer.

The local context module first applies adaptive max pooling to the feature map to introduce the contextual information of the target. By adjusting the output size of the adaptive max pooling, the focus range of this module can be controlled. Then, convolutional layers, ReLU activation functions, and sigmoid activation functions are used to obtain the weights of the contextual information. Finally, the contextual weights are multiplied with the feature map to extract the local contextual information, enhancing the expressive power of the input feature map for target discriminative. This module can be represented as follows:

$$L(X) = \sigma(W_2(\delta(W_1(AP(X))))) \tag{1}$$

where $X \in \mathbb{R}^{W \times H}$ represents the input feature map, $AP(\cdot)$ denotes the adaptive max pooling function, $W_1(\cdot)$ and $W_2(\cdot)$ represent the convolutional layers, $\delta(\cdot)$ is the ReLU activation function, $\sigma(\cdot)$ is the sigmoid activation function, and $L(\cdot)$ represents the local context module.

The local contrast extraction module sequentially calculates the local mean, variance, and standard deviation of the feature map. First, the local mean is computed using the average pooling function. Then, the local variance is calculated using the local mean and the average pooling function. On this basis, the standard deviation is computed. Finally, a convolutional layer is applied to obtain the local contrast. The purpose of this module is to extract the local contrast by introducing manually designed features to enhance the contrast between the target and its surroundings. It captures the grayscale differences between the target and its surrounding pixels, helping the network better differentiate between the target and the background. This module can be represented as follows:

$$C(X) = W(STD(X)) \tag{2}$$

where $X \in \mathbb{R}^{W \times H}$ represents the input feature map, $STD(\cdot)$ represents the standard deviation, $W(\cdot)$ represents the convolution layer, and $C(\cdot)$ represents the local contrast module.

The local attention mechanism is inspired by the SE attention mechanism [47]. The SE attention mechanism primarily focuses on learning channel-wise relationships through global average pooling and fully connected layers. However, it mainly emphasizes channel weights, which results in the loss of local detail information and insensitivity to scale variations of the targets. In this paper, improvements are made based on the SE attention mechanism. Firstly, two convolutional layers are used to extract different representations of

the input feature map, in order to further enhance the feature expressive capacity and focus more on the relationships between local features. Then, the results obtained from these two convolutional layers are multiplied and an attention weight of the input feature is calculated using a fully connected layer. Compared to the SE attention mechanism, the proposed local attention mechanism has fewer parameters. Finally, the attention weight is multiplied with the input feature map to obtain the processed result using the local attention mechanism. The purpose of this module is to compute the attention weights of the input feature map in a learned manner, perform weighted fusion on the original feature map, and enhance the network's focus on local features. This module can highlight important local features. By introducing local feature attention and multi-channel feature interaction, this design enables the local feature extraction module to possess stronger expressive capacity and local feature separation ability when addressing dense object detection problems. This module can be represented as follows:

$$A(X) = X \otimes \delta(W_3(\delta(W_1(X)) \otimes \delta(W_2(X)))) \tag{3}$$

where $X \in \mathbb{R}^{W \times H}$ represents the input feature map; $\delta(\cdot)$ is the ReLU activation function; $W_1(\cdot)$, $W_2(\cdot)$, and $W_3(\cdot)$ represent the convolution layers; $\otimes$ denotes the element-wise multiplication; and $A(\cdot)$ represents the local attention module.

The local feature extraction module can be obtained by adding the output feature map of the local contrast module and the output feature map of the local attention module, and then multiplying the result with the output feature map of the local context module. Finally, the channel number can be adjusted through a convolutional layer. By adjusting the output size of the adaptive max pooling in the local context module and the convolutional layer size and number in the local context, local contrast, and local attention modules, the scope of the local feature extraction module's effect can be controlled, achieving multi-scale feature extraction. This module can be represented as follows:

$$LF(X) = (C(X) + A(X)) \otimes L(X) \tag{4}$$

where $X \in \mathbb{R}^{W \times H}$ represents the input feature map; $LF(X)$, $L(X)$, $C(X)$, and $A(X)$ respectively represent the output feature map of the local feature extraction module, the local contrast module, the local attention module, and the local context module.

To address the issue of significant scale variations in infrared small targets, a multi-scale local feature pyramid module was designed, which consists of multiple stacked local feature extraction modules. For the input infrared image $I \in \mathbb{R}^{W \times H}$, the backbone network undergoes three down-sampling steps during the feature extraction process, resulting in three feature maps of different sizes: $I_1 \in \mathbb{R}^{(W/2) \times (H/2)}$, $I_2 \in \mathbb{R}^{(W/4) \times (H/4)}$, and $I_3 \in \mathbb{R}^{(W/8) \times (H/8)}$. By analyzing these three feature maps, it can be observed that both large and small targets' features are preserved. However, as the feature map size decreases, the features of small targets in the feature maps may be lost to some extent. To tackle this problem, this paper proposes three different multi-scale local feature pyramid modules, each designed specifically for the feature maps of the three different sizes.

The multi-scale local feature pyramid module is designed to address the significant variations in target sizes within the feature maps. In order to extract semantic features of the targets while preserving features of different sizes, this paper proposes stacking three local feature extraction modules of different scales to form the multi-scale local feature pyramid module. The architecture of the multi-scale local feature pyramid module for the feature map $I_1 \in \mathbb{R}^{(W/2) \times (H/2)}$ is shown in Figure 4.

The convolutional kernel sizes for the three local feature extraction modules are $3 \times 3$, $5 \times 5$, and $7 \times 7$. The output sizes of the adaptive max pooling are $25 \times 25$, $19 \times 19$, and $15 \times 15$. Taking the example of the convolutional kernel sizes and the output sizes of the adaptive max pooling, this module focuses on the targets in the feature map and introduces context information slightly larger than the range through adaptive max pooling. The reasons for choosing different convolutional kernel sizes and adaptive max pooling output

sizes are as follows. Through an experimental analysis, it was found that in the feature maps $I_1$, there was significant variation in the target sizes. Therefore, we chose three different kernel sizes that matched the target sizes and calculated the adaptive max pooling output size to be slightly larger than the target size to capture contextual information around the target. The results of the three local feature extraction modules at these three scales are summed together to obtain the final output.

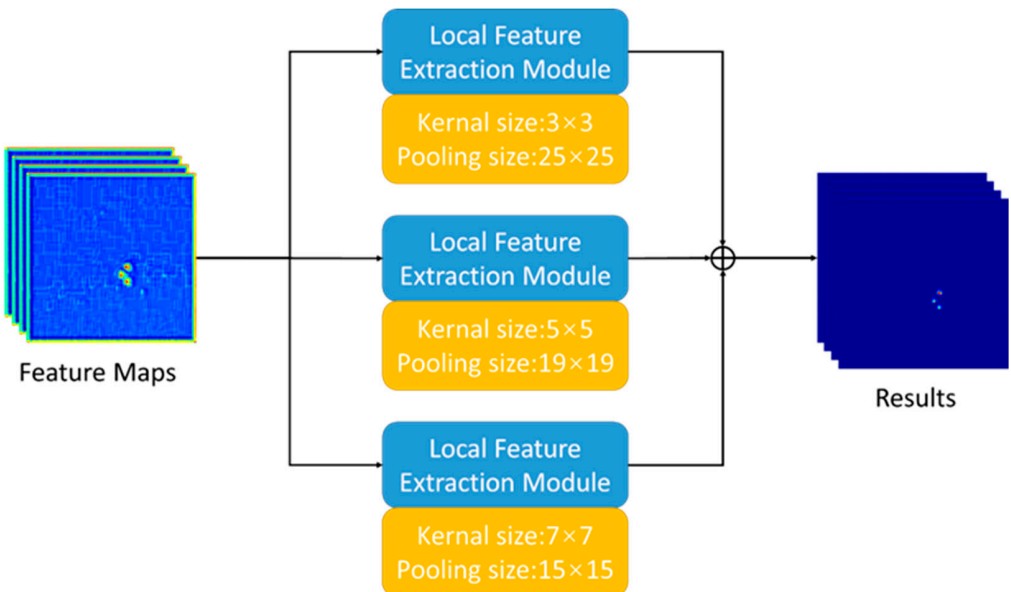

**Figure 4.** Multi-scale local feature pyramid module. The multi-scale local feature pyramid module consists of three local feature extraction modules with different kernel and pooling sizes.

For the feature map $I_2 \in \mathbb{R}^{(W/4) \times (H/4)}$ with a moderate range of target size variations, a multi-scale local feature pyramid module was designed by combining two local feature extraction modules. The convolutional kernel sizes for these modules are $3 \times 3$ and $5 \times 5$, and the output sizes of the adaptive max pooling are $13 \times 13$ and $9 \times 9$. Through an experimental analysis, it was observed that as the feature map size decreases, the range of target size variations also decreases. Therefore, we opted for only two kernel sizes and did not use a $7 \times 7$ kernel size. Through calculations, we determined the adaptive max pooling output size to be slightly larger than the target size to capture contextual information around the target. For the feature map $I_3 \in \mathbb{R}^{(W/8) \times (H/8)}$ with the smallest range of target size variations, only one local feature extraction module is used. The convolutional kernel size for this module is $3 \times 3$, and the output size of the adaptive max pooling is $7 \times 7$. As the target size in the deepest feature map is very small, we retained only the $3 \times 3$ convolutional kernel and selected an adaptive max pooling output size that aligned with it.

2.2.3. Multi-Scale Feature Fusion Module

The structure of the multi-scale feature fusion module is shown in Figure 5.

This module can be represented as follows:

$$P = W(\text{Concat}(M_1, M_2, M_3)) \tag{5}$$

where $W(\cdot)$ represents the convolutional layer and $\text{Concat}(\cdot)$ represents the concatenation operation. $M_1$, $M_2$, and $M_3$ can be represented as follows:

$$M_1 = M(X_1) \tag{6}$$

$$M_2 = F(M_1, \text{Up}(M(X_2))) \tag{7}$$

$$M_3 = F(M_1, Up(F(M(X_2), Up(M(X_3))))) \tag{8}$$

where $M(X_1)$, $M(X_2)$, and $M(X_3)$ represent the outputs of the multi-scale local feature pyramid module for feature maps $X_1 \in \mathbb{R}^{(W/2) \times (H/2)}$, $X_2 \in \mathbb{R}^{(W/4) \times (H/4)}$, and $X_3 \in \mathbb{R}^{(W/8) \times (H/8)}$, respectively; $F(\cdot)$ donates the asymmetric fusion module (AFM) [41], $Up(\cdot)$ represents the up-sampling operation, and P is the final detection result.

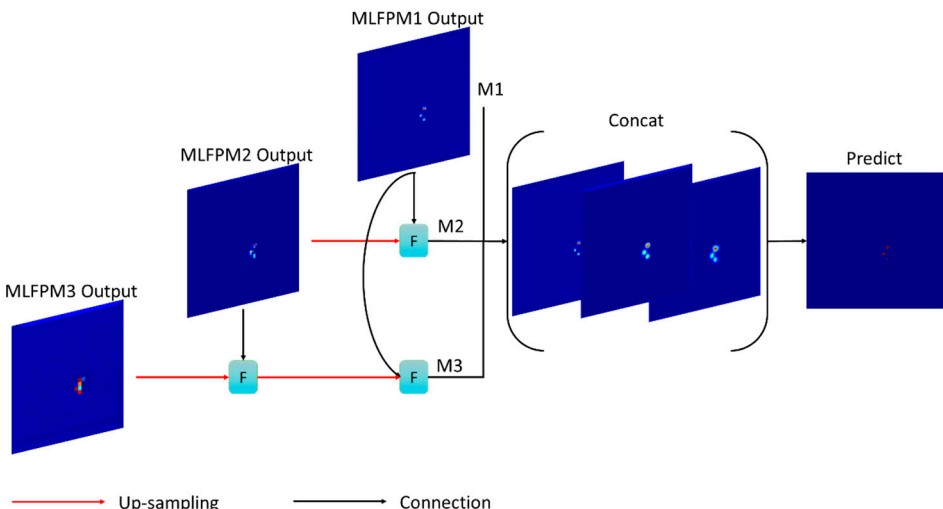

**Figure 5.** Multi-scale feature fusion module. The multi-scale feature fusion module integrates features from different scales obtained at three different stages. The MLFPM1 output, MLFPM2 output, and MLFPM3 output in the figure are the results of processing performed by MLFPMs, while M1, M2, and M3 represent the respective fusion results.

MFFM takes $M(X_1)$, $M(X_2)$, and $M(X_3)$ as inputs. For $M(X_3)$, it first undergoes up-sampling, followed by feature fusion with $M(X_2)$. The fused result is then up-sampled again and further fused with $M(X_1)$ to obtain M3. For $M(X_2)$, it undergoes up-sampling and is fused with $M(X_2)$ itself to generate M2. $M(X_1)$ remains unchanged and is directly used as M1. These M1, M2, and M3 feature maps are concatenated, and a convolutional operation is applied to obtain the final detection result.

## 3. Results

### 3.1. Datasets

#### 3.1.1. SIRST

SIRST extracts 427 representative images from hundreds of real infrared video sequences, containing a total of 480 targets. Additionally, due to the limited availability of infrared sequences, SIRST includes infrared images at wavelengths of not only short and mid-waves but also at a wavelength of 950 nm. Each target is confirmed by observing the motion sequences to ensure it is a real target rather than pixel-level noise. The size of each image is $256 \times 256$ pixels. The dataset provides high-quality annotations. In this dataset, small targets have five annotation forms that adapt to different detection models and support various tasks: image classification, instance segmentation, bounding box regression, semantic segmentation, and instance point recognition. Following the dataset's provided partitioning, the dataset was divided into 342 images for the training set and the remaining 86 images for the test set. Figure 6 displays some representative images from SIRST. Many targets in infrared images are highly blurred and concealed within complex backgrounds. Detecting them is not an easy task, even for humans, as it requires a high level of semantic understanding within the overall scene and focused searching.

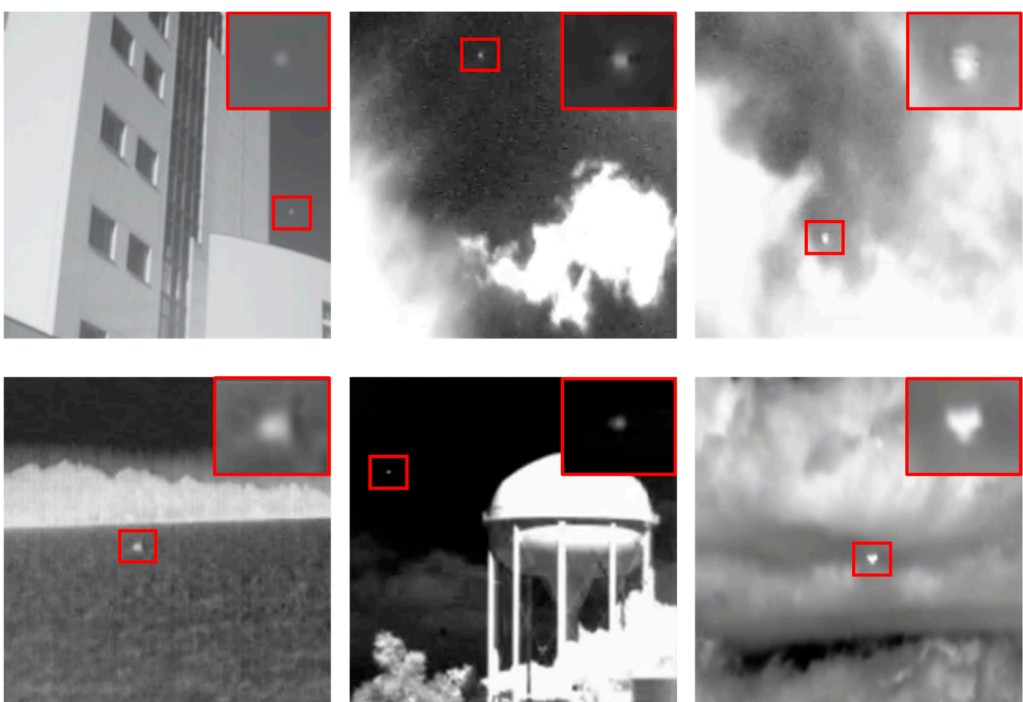

**Figure 6.** Representative images for SIRST. The targets are identified with red boxes.

### 3.1.2. MDFA

MDFA comprises 10,078 images, with each image's size being $128 \times 128$ pixels. Following the dataset's provided partitioning, the dataset was divided into 9987 images for the training set, with the remaining 100 images being used for the test set. Figure 7 displays some representative images from MDFA. Not all images in the MDFA dataset are real infrared images. Due to the lack of infrared small target dataset, MDFA collects real infrared images and synthesizes new infrared images, collects infrared high-resolution natural scene images from the Internet, and crops these images in different areas to form different backgrounds. Then, the small target separated from the real infrared image and the small target object synthesized by the two-dimensional Gaussian function are superimposed on the obtained background to form a new image. The approach for separating small targets from real images involves directly segmenting the targets based on annotated masks. The grayscale distribution of the targets is generated through the two-dimensional Gaussian function proposed in Equation (9):

$$s(x,y) = \exp\{-[((\cos(\alpha)x - \sin(\alpha)y)/\ w_x)^2 + ((\cos(\alpha)y + \sin(\alpha)x)/\ w_y)^2]\} \quad (9)$$

where $w_x$ and $w_y$ represent the dimensions of the Gaussian distribution in two vertical directions, with values ranging from 1 to 3; $\alpha$ determines the distribution direction, with values ranging from 0 to $\pi$; $s(x,y)$ represents the grayscale value at position $(x,y)$ in the image.

### 3.2. Evaluation Metric

To accurately assess the detection capabilities of different methods, this paper utilizes classic semantic segmentation evaluation metrics, including the precision, recall, F-measure, mean intersection over union (mIoU), and area under the curve (AUC). Higher precision, recall, F-measure, mIoU, and AUC values indicate stronger detection abilities for the methods.

The selection of precision as one of our evaluation metrics was driven by the paramount importance of accuracy in infrared small target detection. A high precision value signifies the model's reduced likelihood of misclassifying non-target objects, a critical factor in tasks such as target tracking and military applications. The recall assesses the model's capability to detect all targets within the dataset. In the context of infrared small target

detection, where targets can be exceedingly small and scattered, the goal is to ensure the model minimizes missed detections. High recall signifies the model's effectiveness in detecting the majority of small targets, thereby mitigating the likelihood of false negatives. Although the mIoU is typically employed in image segmentation tasks, it also holds significance in object detection. It quantifies the degree of overlap between the model's detection outcomes and the actual target regions, providing an assessment of the target localization quality. In the realm of small target detection, precise target localization is pivotal, and the mIoU provides insights into the spatial accuracy of the model's predictions. The F-measure, as the harmonic mean of the precision and recall, provides a balanced assessment of a model's accuracy and recall. This is useful for balancing false positives and false negatives. In small target detection, consideration must be given to both false alarms and missed detections, rendering the F-measure a comprehensive performance metric. The AUC assesses the model's capability to differentiate between positives and negatives across various thresholds. In the realm of small target detection, the AUC evaluates the model's comprehensive ability to distinguish between targets and non-targets, serving as an indicator of its discriminative power.

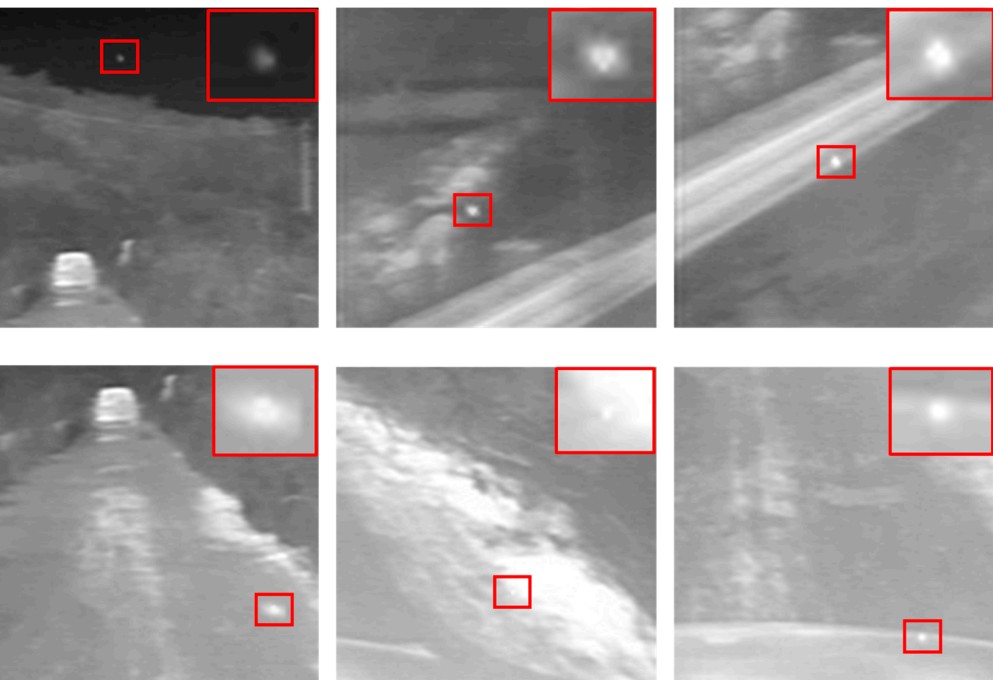

**Figure 7.** Representative images for MDFA. The targets are identified with red boxes.

The selection of these metrics is deliberate, aiming to provide a thorough evaluation of infrared small target detection methods. These metrics take into account key aspects, including the accuracy, recall, spatial precision, overall model performance, and discriminative capability.

*3.3. Implementation Details*

The entire method was implemented using the PyTorch framework. The choice of the PyTorch framework was primarily driven by its widespread adoption and its reputation for flexibility and ease of use in the deep learning community. PyTorch provides a rich set of tools for developing and training neural networks, making it a suitable choice for implementing and experimenting with complex deep learning models such as the one proposed in our study. In this paper, ResNet-18 is used as the backbone network and is retrained along with the other parts of the network using the Stochastic Gradient Descent (SGD) optimizer. The momentum and weight decay are set to 0.9 and 0.0004, respectively. The initial learning rate is set to 0.05 and is multiplied by $(1 - \text{iter}/total\_iter)^{0.9}$ after each

iteration. The SoftIoU loss function is employed. For SIRST, the method is trained for 400 epochs, while for MDFA, it is trained for 5 epochs. The training process is performed using an Nvidia Titan XP GPU with 12GB of memory.

### 3.4. Comparisons with State-of-the-Art Methods

To further validate the effectiveness of our proposed method, a comparison was conducted with state-of-the-art methods, including traditional methods (LCM [25], RLCM [26], PSTNN [48], MPCM [49], IPI [33]) and deep-learning-based methods (MDvsFA [16], ACM [17], AGPCNet [41]). The hyper-parameters for all methods were set according to the respective papers or publicly available code, and these hyper-parameters are shown in Table 1. In Tables 2 and 3, quantitative evaluations of different methods were conducted on SIRST and MDFA. The maximum value in each column is highlighted in bold black font. ACM and AGPCNet were retrained using the dataset segmentation approach described in this paper.

**Table 1.** Hyper-parameter settings for state-of-the-art methods.

| Methods | Hyper-Parameters Settings |
|---------|---------------------------|
| LCM | Filter radius: 1, 2, 3, 4 |
| RLCM | Filter radius: 1, 2, 3, 4 |
| PSTNN | Patch size: $40 \times 40$, Slide step: 40, $\lambda = 0.7/\sqrt{\max(n_1, n_2)}$ |
| MPCM | Filter radius: 1, 2, 3, 4 |
| IPI | Patch size: $50 \times 50$, Slide step: 10, $\lambda = 1/\sqrt{\max(m, n)}$ |

**Table 2.** Comparison with state-of-the-art methods on SIRST. The maximum value in each column is highlighted in bold black font.

| Methods | Precision | Recall | mIoU | F-Measure | AUC |
|---------|-----------|--------|------|-----------|-----|
| LCM | 0.0241 | 0.9087 | 0.2015 | 0.0469 | 0.7208 |
| RLCM | 0.0111 | **0.9164** | 0.0094 | 0.0219 | 0.9509 |
| PSTNN | 0.7893 | 0.5198 | 0.4262 | 0.6268 | 0.7131 |
| MPCM | 0.0052 | 0.8550 | 0.0046 | 0.0102 | 0.8986 |
| IPI | 0.7541 | 0.5749 | 0.6216 | 0.6524 | 0.8485 |
| MDvsFA | — | — | — | — | — |
| ACM | 0.6262 | 0.7531 | 0.5196 | 0.6838 | 0.9053 |
| AGPCNet | **0.6858** | 0.8424 | 0.6078 | 0.7561 | 0.9321 |
| Ours | 0.6757 | 0.9118 | **0.6343** | **0.7762** | **0.9577** |

**Table 3.** Comparison with state-of-the-art methods on MDFA. The maximum value in each column is highlighted in bold black font.

| Methods | Precision | Recall | mIoU | F-Measure | AUC |
|---------|-----------|--------|------|-----------|-----|
| LCM | 0.0192 | 0.7538 | 0.4085 | 0.0375 | **0.9979** |
| RLCM | 0.0047 | **0.9359** | 0.0276 | 0.0094 | 0.9360 |
| PSTNN | 0.4369 | 0.4996 | 0.3304 | 0.4661 | 0.7752 |
| MPCM | 0.0023 | 0.8859 | 0.0086 | 0.0046 | 0.7850 |
| IPI | 0.4674 | 0.5471 | 0.2836 | 0.5041 | 0.6418 |
| MDvsFA | **0.6600** | 0.5400 | — | 0.6000 | 0.9100 |
| ACM | 0.4615 | 0.7177 | 0.3906 | 0.5617 | 0.9029 |
| AGPCNet | 0.5490 | 0.7231 | 0.4537 | 0.6242 | 0.8814 |
| Ours | 0.5992 | 0.6705 | **0.4629** | **0.6328** | 0.8382 |

### 3.4.1. Results on SIRST

From Table 2, it can be observed that traditional methods often struggle to balance the precision and recall. LCM, RLCM, and MPCM exhibit high recall but low precision,

indicating a high number of false positives in the detection results, while PSTNN and IPI show high precision but low recall, indicating instances of missed detection. This is because the traditional methods primarily focus on the local features of the targets and have limited background suppression capability, making them susceptible to the influence of complex backgrounds. Deep-learning-based methods show improved recall while maintaining high precision. As shown in Table 1, our proposed method achieved a 67.57% precision rate, which is higher than many advanced methods. Our proposed method achieved a 91.18% recall rate, which is second only to the RLCM. Our proposed method achieved the maximum values for the mIoU, F-measure, and AUC. This indicates that compared to other state-of-the-art methods, our proposed method effectively suppresses the background and accurately detects the targets.

In the case where the precision and recall show higher values for other methods, this discrepancy arises due to differences between traditional and deep-learning-based methods. Traditional methods tend to excel in scenarios with straightforward backgrounds because they primarily focus on the local features of the targets. However, they often struggle with maintaining both precision and recall when dealing with complex backgrounds. Conversely, deep-learning-based methods leverage the powerful capabilities of neural networks to enhance the recall while maintaining competitive precision rates, effectively striking a balance between the two aspects. To comprehensively assess the performance of our method, we conducted further evaluations using the F-measure, which considers the balance between the precision and recall. Therefore, even though our proposed method did not achieve the highest values for both precision and recall, having the highest F-measure still indicates its exceptional performance. This is because the F-measure's calculation method takes into account the trade-off between the precision and recall, enabling a more comprehensive evaluation of our method's overall performance.

### 3.4.2. Results on MDFA

As shown in Table 3, our proposed method achieved a 59.92% precision rate, which is second only to the MDvsFA. Our proposed method achieved a 67.05% recall rate, which is higher than many advanced methods. Our proposed method achieved the maximum values for the mIoU and F-measure.

Consistent with the comparison results on the SIRST dataset, our proposed method also did not achieve the highest values for precision and recall individually. However, it demonstrated comparatively high values for both precision and recall. Notably, it achieved the highest F-measure value, which underscores the superiority of our proposed approach. For the LCM and RLCM, they achieved high detection rates in scenarios with relatively high false alarm rates, consequently yielding high AUC values to approach zero, subsequently leading to F-measure values close to zero as well. Therefore, it is crucial to consider multiple metrics when assessing the method performance.

In Figures 8 and 9, two representative infrared images were selected to compare the detection results of the eight methods. The target locations are marked with red bounding boxes, while missed detections and false positives are marked with yellow bounding boxes. By observing Figure 8, it can be noted that traditional methods can detect the objects but are often affected by complex background interference, resulting in a large number of false positives. Large numbers of false alarms can be observed in the detection results of the LCM and MPCM algorithms. In the detection results of the RLCM, PSTNN, and IPI algorithms, false alarms occur in locally highlighted areas. ACM and AGPCNet exhibit instances of missed detections. The proposed method in this paper achieves the best detection results, accurately detecting the targets without any false positives. The proposed method achieves the best detection results by effectively suppressing the background and accurately detecting the targets.

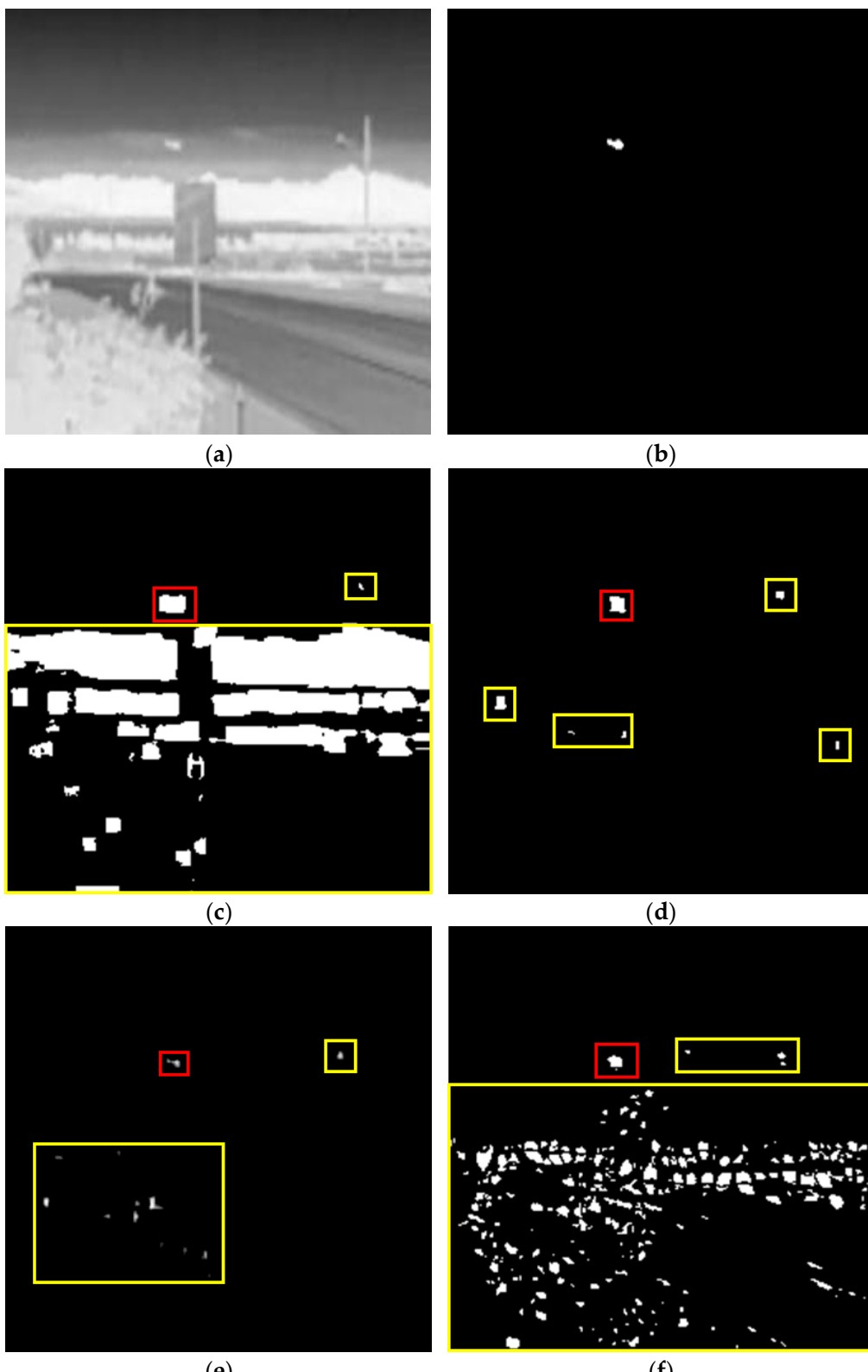

**Figure 8.** *Cont.*

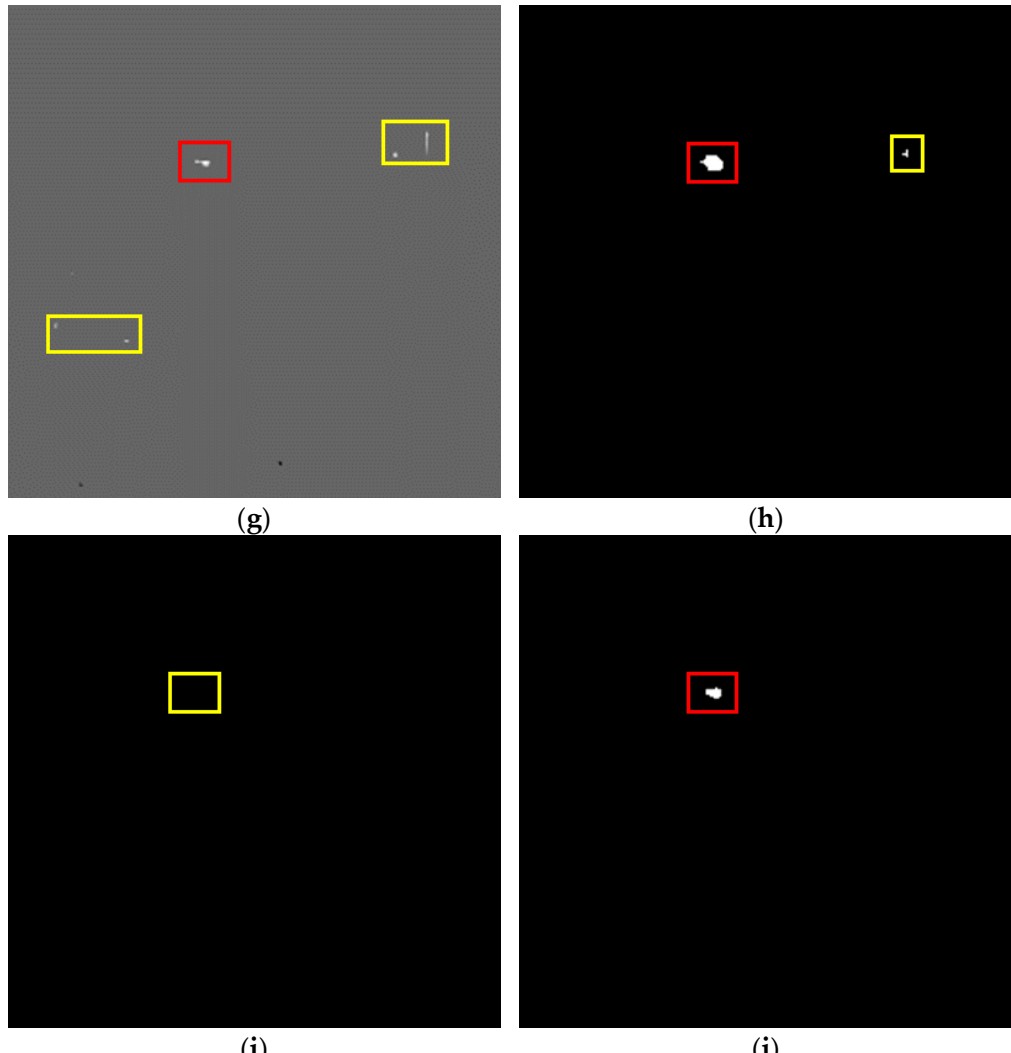

**Figure 8.** Detection results for infrared scene 1: (**a**) input image; (**b**) ground truth; (**c**) detection result for LCM; (**d**) detection result for RLCM; (**e**) detection result for PSTNN; (**f**) detection result for MPCM; (**g**) detection result for IPI; (**h**) detection result for ACM; (**i**) detection result for AGPCNet; (**j**) detection result for our proposed method. The target locations are marked with red bounding boxes, while missed detections and false positives are marked with yellow bounding boxes.

As shown in Figure 9, the detection results for the traditional methods also show significant numbers of false positives. Regarding the detection results for the traditional methods, LCM, PSTNN, MPCM, and IPI are all capable of detecting the targets, although these methods also suffer from false alarms. RLCM exhibits instances of missed detection for densely packed targets. Although ACM and AGPCNet exhibit strong background suppression, they still suffer from false positives and missed detections. The proposed method in this paper achieves the best detection results, accurately detecting the targets without any false positives. This indicates that the proposed method in this paper is capable of effectively addressing scenarios with densely packed targets.

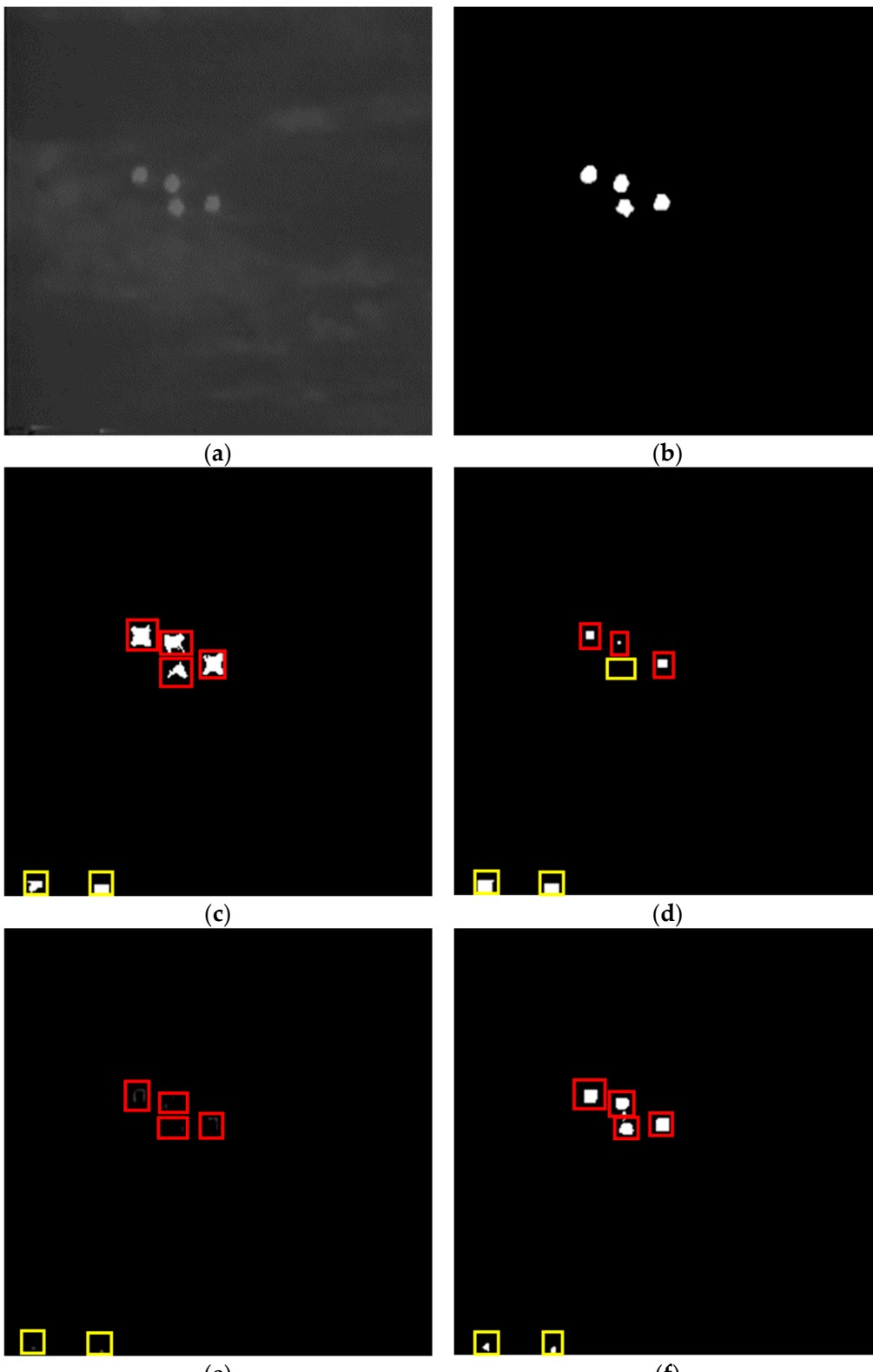

**Figure 9.** *Cont.*

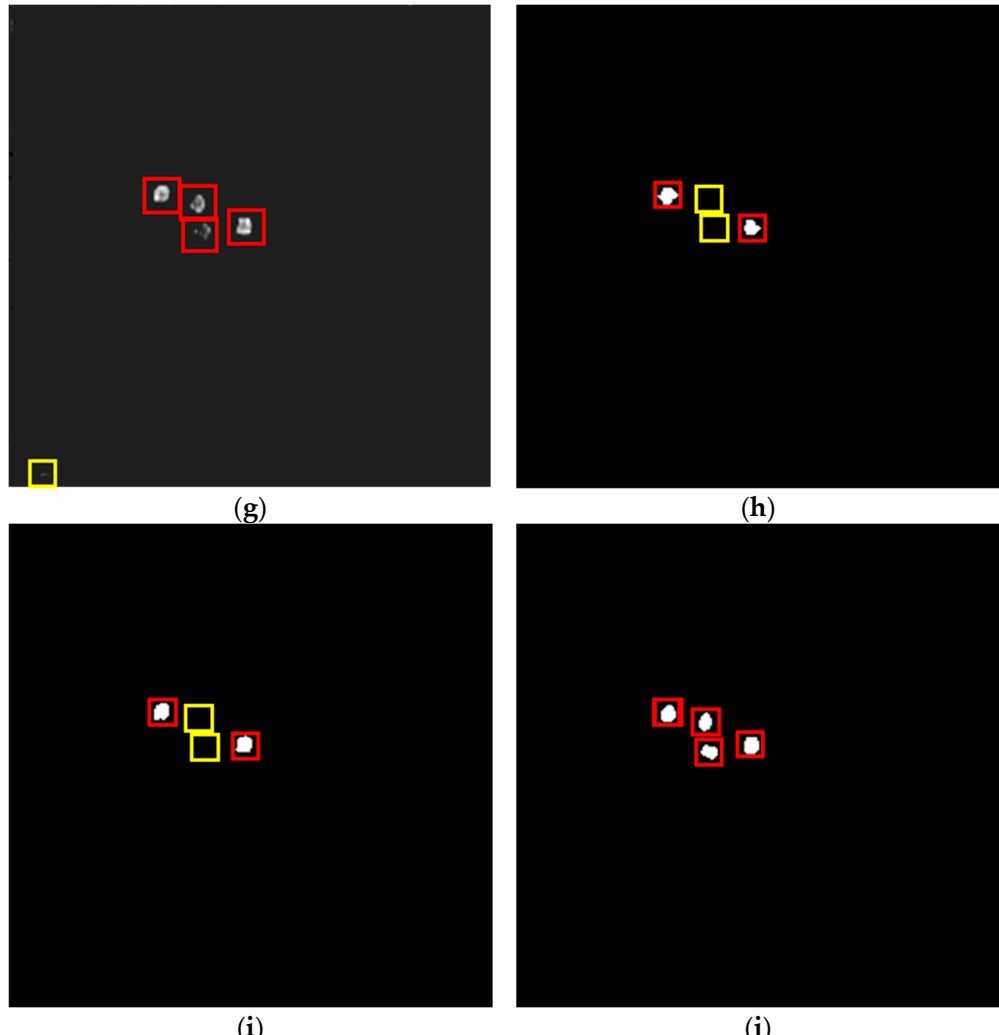

**Figure 9.** Detection results for infrared scene 2: (**a**) input image; (**b**) ground truth; (**c**) detection result for LCM; (**d**) detection result for RLCM; (**e**) detection result for PSTNN; (**f**) detection result for MPCM; (**g**) detection result for IPI; (**h**) detection result for ACM; (**i**) detection result for AGPCNet; (**j**) detection result for our proposed method. The target locations are marked with red bounding boxes, while missed detections and false positives are marked with yellow bounding boxes.

### 3.4.3. Running Time

The average running times for the state-of-the-art methods and proposed method were calculated on images measuring $256 \times 256$ pixels, as shown in Table 4. Among the traditional methods, except for RLCM, the rest of the methods generally outperform the deep-learning-based methods in terms of the running time. The proposed method does not exhibit an advantage in terms of running speed.

**Table 4.** A running time analysis on images measuring $256 \times 256$ pixels.

| Methods | Running Time on GPU/s |
| --- | --- |
| LCM | 0.163 |
| RLCM | 8.987 |
| PSTNN | 0.272 |
| MPCM | 0.093 |
| IPI | 23.913 |
| ACM | 0.077 |
| AGPCNet | 0.423 |
| Ours | 0.947 |

## 4. Discussion

### 4.1. Ablation Study

To demonstrate the effectiveness of the proposed model, ablation studies were conducted on the SIRST and MDFA. The results of the ablation experiments are presented in Table 1, with the maximum value for each metric highlighted in bold black font. It can be seen from Table 5 that the proposed method with both the MLFPM and MFFM modules achieved a significant improvement. For SIRST, when both MLFPM and MFFM were added, all four metrics showed improvements, with the recall, mIoU, and F-measure values being the highest and the precision value being close to the maximum. For MDFA, when both MLFPM and MFFM were added, all four metrics showed improvements, with the precision, mIoU, and F-measure values being the highest.

**Table 5.** Ablation study on the whole network. The maximum value in each column is highlighted in bold black font.

| Dataset | Backbone | MLFPM | MFFM | Precision | Recall | mIoU | F-Measure | AUC |
|---------|----------|-------|------|-----------|--------|------|-----------|-----|
| SIRST | | | | 0.6437 | 0.8729 | 0.5886 | 0.7410 | 0.9252 |
| | | √ | | **0.6777** | 0.8818 | 0.6213 | 0.7664 | 0.9526 |
| | | | √ | 0.6658 | 0.8504 | 0.5960 | 0.7469 | 0.9319 |
| | ResNet-18 | √ | √ | 0.6757 | **0.9118** | **0.6343** | **0.7762** | **0.9577** |
| MDFA | | | | 0.5674 | 0.6574 | 0.4379 | 0.6091 | 0.8454 |
| | | √ | | 0.5439 | **0.7231** | 0.4502 | 0.6208 | 0.8529 |
| | | | √ | 0.5630 | 0.7185 | 0.4613 | 0.6313 | **0.8833** |
| | | √ | √ | **0.5992** | 0.6705 | **0.4629** | **0.6328** | 0.8382 |

#### 4.1.1. Effect of the Proposed MLFPM

The results of the ablation study on the whole network are shown in Table 5. The results of the ablation study on SIRST and MDFA indicate that the detection performance is improved by adding MLFPM to the backbone network. For SIRST, adding MLFPM resulted in improvements in all five metrics. The precison, recall, mIoU, F-measure and AUC increased by 3.40%, 0.89%, 3.27%, 2.54%, and 2.74%, respectively. For MDFA, adding MLFPM led to improvements in the recall, mIoU, F-measure, and AUC. The recall, mIoU, F-measure, and AUC increased by 6.57%, 1.23%, 1.17%, and 0.75%, respectively. However, the precison decreased by 2.35%. The results prove the effectiveness of the MLFPM module. The design of the proposed MLFPM offers an effective approach to address the challenge of detecting infrared small targets at various scales. Given the broader context of infrared small target detection, where targets can exhibit significant variations in size, the existing methods often struggle to deliver satisfactory detection performance for both small and large targets. This module processes multiple layers of feature maps, ensuring the extraction and preservation of target features at different scales.

#### 4.1.2. Effect of the Proposed MFFM

As shown in Table 5, adding the MFFM resulted in improvements in the precision, mIoU, F-measure, and AUC values for SIRST. The precision, mIoU, F-measure, and AUC values increased by 2.21%, 0.74%, 0.59%, and 0.67%, respectively. However, the recall decreased by 2.25%. For MDFA, adding MFFM led to improvements in recall, mIoU, F-measure, and AUC. The recall, mIoU, F-measure, and AUC increased by 6.11%, 2.34%, 2.22%, and 3.79%, respectively. However, the precison decreased by 0.44%. The results prove the effectiveness of the MFFM module. The design of the proposed MFFM presents an effective approach to address the challenges in dense infrared small target detection. Within the broader context of infrared small target detection, the existing methods often struggle to appropriately fuse the semantic information from high-level feature maps with the spatial information from low-level feature maps, leading to inaccuracies in detecting dense targets. This issue typically manifests as the omission of targets located in the middle

of dense targets. The MFFM module achieves the preservation of valuable information by reusing both high-level and low-level features.

### 4.1.3. Effect of the Three Components in the MLFPM

The results of the ablation study on MLFPM are shown in Table 6. To demonstrate the effectiveness of the local context module, local contrast extraction module, and local attention mechanism module with MLFPM, ablation studies were conducted on SIRST and MDFA. The results are presented in Table 6. On SIRST, the network that incorporates all three modules simultaneously achieved the highest values across all metrics. On MDFA, the network that incorporates all three modules simultaneously achieved the highest values for the recall, mIoU, F-measure, and AUC. These results indicate that incorporating these three modules effectively enhances the detection capabilities of the network.

**Table 6.** Ablation study on MLFPM. The maximum value in each column is highlighted in bold black font.

| Dataset | Local Context | Local Contract Extraction | Local Attention | Precision | Recall | mIoU | F-Measure | AUC |
|---------|---------------|---------------------------|-----------------|-----------|--------|------|-----------|-----|
| SIRST | | √ | √ | 0.6609 | 0.9067 | 0.6188 | 0.7645 | 0.9574 |
| | √ | | √ | 0.6597 | 0.8649 | 0.5981 | 0.7485 | 0.9283 |
| | √ | √ | | 0.6554 | 0.8752 | 0.5994 | 0.7495 | 0.9437 |
| | √ | √ | √ | **0.6757** | **0.9118** | **0.6343** | **0.7762** | **0.9577** |
| MDFA | | √ | √ | **0.6391** | 0.6164 | 0.4573 | 0.6276 | 0.7762 |
| | √ | | √ | 0.5915 | 0.5599 | 0.4038 | 0.5753 | 0.4902 |
| | √ | √ | | 0.5997 | 0.6466 | 0.4516 | 0.6223 | 0.7640 |
| | √ | √ | √ | 0.5992 | **0.6705** | **0.4629** | **0.6328** | **0.8382** |

### 4.2. Advantages and Limitations

As demonstrated in the comparison presented in the Results section and the ablation study discussed in the Discussion section, the proposed MLFPM effectively enhances the extraction of local features and contextual information from feature maps. It significantly contributes to boosting the semantic information on both high-level and low-level features. The introduced MFFM, by reusing high-level and low-level features, harnesses the rich semantic information from low-level feature maps, resulting in improved prediction performance across various metrics as compared to the existing methods. The overall detection results outperform the current methods.

However, as indicated in Table 4, in the design process for the MLFPM, various multi-scale convolution operations were employed on each layer to further enhance the feature extraction. Given the larger dimensions of shallow feature maps, applying diverse multi-scale convolutions increases the computational complexity. Consequently, this leads to a prolonged running time and strained computational resources. This challenge represents an area of focus for future research.

### 5. Conclusions

This paper presents an infrared small target detection method based on multi-stage and multi-scale local feature fusion. The proposed method consists of two key modules: the multi-scale local feature pyramid module and the multi-scale feature fusion module. MLFPM aims to further enhance the semantic features of feature maps at different scales, suppress background interference, and address the problem of target scale variation. MFFM integrates feature maps of different scales, effectively solving the issue of dense targets being difficult to separate. Both ablation studies and comparisons with state-of-the-art methods were conducted on SIRST and MDFA. The former validated the effectiveness of each component in the constructed model, while the latter demonstrated the superiority

of the proposed method over similar approaches. On SIRST, an mIoU of 63.43% and F-measure of 77.62% were achieved. On MDFA, an mIoU of 46.29%and F-measure of 63.28% were obtained. By introducing these two modules, the model's detection performance for multi-scale and dense targets was enhanced. However, it is essential to acknowledge that the complex MLFPM design has implications for the computational efficiency, potentially impacting the processing speed. Furthermore, there is room for improvement in the detection rate and precision during infrared small target detection. Our future work will focus on optimizing the network, reducing the computational load, and exploring better methods for multi-scale and densely distributed infrared small target detection.

**Author Contributions:** Conceptualization, Y.W. and Y.T.; methodology, Y.W.; software, Y.W.; validation, Y.W., Y.T. and Y.X.; formal analysis, Y.W and Y.T.; investigation, Y.X.; resources, Y.T.; data curation, Y.W., Y.X. and J.L.; writing—original draft preparation, Y.W.; writing—review and editing, Y.T.; visualization, J.L. and Y.X.; supervision, Y.T.; project administration, Y.W. All authors have read and agreed to the published version of the manuscript.

**Funding:** This research received no external funding.

**Data Availability Statement:** The datasets used in this study are available on request from the corresponding author.

**Conflicts of Interest:** The authors declare no conflict of interest.

## Abbreviations

The following abbreviations are used in this manuscript:

| | |
|---|---|
| AUC | Area under the curve |
| FPN | Feature pyramid network |
| mIoU | mean intersection over union |
| MLFPM | Multi-scale local feature pyramid module |
| MFFM | Multi-scale feature fusion module |
| RoI | Region of interest |

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
