# Peer review of "Multi-Stage Multi-Scale Local Feature Fusion for Infrared Small Target Detection"

_remotesensing, doi:10.3390/rs15184506_

Round 1

Reviewer 1 Report

Multi-Stage Multi-Scale Local Feature Fusion for Infrared 2 Small Target Detection

1.     The abstract and conclusion need to technically strengthen. Please rewrite it.

2.     In abstract use full forms first time, acronyms cannot be used directly.

3.     The article is written from the third person’s perspectives hence don’t use “I”, “We” and such other words.

4.     The equation must be written in a proper format.

5.     In the equations, integers, and constants used must be elaborate in detail to understand the mathematic concepts thoroughly.

6.     Provide some technical comments on research methods or content

7.     Illustrate what are the manuscript’s strengths and weaknesses;

8.     Overall paper is well-written and seems technically strong after correcting these corrections/suggestions.

-

Reviewer 2 Report

Let me summarize my concerns related to the aritc

Introduction:

The introduction effectively establishes the importance of object detection in various applications and highlights the challenges specific to infrared small target detection. However, it could benefit from more specific contextualization within the field of computer vision and the relevance of the proposed method to existing research. Additionally, a brief discussion on the limitations of traditional methods and the need for novel approaches could enhance the introduction's clarity.

Background and Related Work:

The article briefly touches on two-stage and single-stage object detection algorithms without providing a deeper comparison or analysis of their relevance to the proposed approach. To enhance this section, a more comprehensive overview of existing infrared small target detection methods, both traditional and deep learning-based, would provide a stronger foundation for understanding the proposed contributions.

Proposed Method:

The proposed method's components are well-described, particularly the Multi-scale Local Feature Pyramid Module (MLFPM) and Multi-scale Feature Fusion Module (MFFM). However, some details could be further elaborated. For instance, the specific choices for kernel sizes and output sizes of adaptive max pooling in the MLFPM for different feature map sizes could be explained in more depth. Additionally, while the method's architecture is well-presented, the reasons for selecting certain convolutional kernel sizes, should be included to strengthen the reader's understanding.

In addition, equations should be readible.

Experimental Results:

The article does not explain why the chosen deep learning framework (PyTorch) is appropriate for the task of infrared small target detection. There's also no discussion about why the specific evaluation metrics were chosen over others.

Discussion:

While the contributions of the proposed method are stated, the article could elaborate more on the implications of these contributions in the broader context of infrared small target detection. How does the method address the challenges mentioned in the introduction? How does it advance the field beyond existing approaches? A more comprehensive discussion on these aspects would further solidify the significance of the research.

Conclusion:

The conclusion succinctly summarizes the findings and contributions of the research. However, it could be strengthened by reiterating the specific impact of the proposed method in addressing the challenges of multi-scale and dense target detection in infrared imagery.

Figures and Visuals:

The figures presented in the article effectively illustrate the proposed method's architecture and components. However, figures could benefit from more detailed captions that provide insights into the specific elements being shown.

Citations and References:

The references provided in the article are limited to the existing literature. Including references to other relevant works in the broader field of computer vision and infrared imaging would strengthen the paper's foundation and contextualization.

The writing style is generally formal and appropriate for an academic paper. However, there are some areas where the writing could be further refined for clarity. Sentences such as "For the feature [...] with the smallest range of target size variations, we uses only one local feature extraction module" could be revised for grammatical accuracy, alongside with word selection, i.e. comparison over comperative, lack of the article "The" by the authors before publication.

Reviewer 3 Report

In this paper, Multi-Stage Multi-Scale Local Feature Fusion for Infrared 2 Small Target Detection is proposed. This paper has a clear idea and is well written. There are some flaws in this paper.

1. The formula writing in the text is chaotic and difficult to recognize.

2. In Figure 4. Multi-scale local feature pyramid module, it can be seen that for each scale, only one layer of convolution has been performed. Has the author tried multi-layer convolution, and what are the results?

3. In the experimental section, although the author mentioned “The hyper-parameters for all methods were set according to the respective papers or publicly available code.”, the parameter values of the comparison algorithm still should be provided to more effectively demonstrate the superiority of the proposed algorithm. The comparison algorithm should be in the optimal parameter state.

4. In the experimental section, as shown in Tables 1 and 2, the proposed algorithm is not always optimal. The author should provide appropriate analysis of what is most likely causing the comparative algorithm to be relatively superior.

5. The running time of all testing algorithms should be given.

6. The shortcomings of the proposed algorithm and the direction for future improvement should be given in the conclusion.

Reviewer 4 Report

Implementing the MLFP module may add complexity to the model design. To construct feature pyramids and conduct the necessary actions across multiple sizes, the module requires more processing. This increases the model's computational burden and training time, making it slower and more resource-intensive.

Deep features are often used to collect high-level semantic information, whereas shallow features are used to capture low-level details. Due to the semantic difference between these two sorts of information, combining them might be difficult. It might be challenging to ensure that the fused features accurately represent both the global context and fine-grained details. How do you handle it?

To what extent the proposed attention mechanism handle the fusion features as the deep features often have a higher dimensionality compared to shallow features due to their hierarchical nature. Fusing features from different dimensions can lead to compatibility issues and require careful design of fusion mechanisms

Can you spot on the difference between Multi scale local feature pyramid module and Multi scale guided feature extraction?

The Formula text presented is not in proper font and alignment.

In Line 371 can you provide in detail about " The small target separated from the real infrared image and the small target object synthesized by the two-dimensional Gaussian function are superimposed on the obtained background to form a new image."

The quality of English is Fine.

But the Description to the formula used in the article is not complete.

Round 2

Reviewer 3 Report

The paper has been revised according to the revision suggestions.